# A Cooperative Hunting Method for Multi-USV Based on the A* Algorithm in an Environment with Obstacles

**DOI:** 10.3390/s23167058

**Published:** 2023-08-09

**Authors:** Zhihao Chen, Zhiyao Zhao, Jiping Xu, Xiaoyi Wang, Yang Lu, Jiabin Yu

**Affiliations:** 1School of Artificial Intelligence, Beijing Technology and Business University, Beijing 100048, China; 2130062048@st.btbu.edu.cn (Z.C.); zhaozy@btbu.edu.cn (Z.Z.); xujp@th.btbu.edu.cn (J.X.); wangxy@btbu.edu.cn (X.W.); 2230602063@st.btbu.edu.cn (Y.L.); 2China Food Flavor and Nutrition Health Innovation Center, Beijing Technology and Business University, Beijing 100048, China; 3Laboratory for Intelligent Environmental Protection, Beijing Technology and Business University, Beijing 100048, China; 4School of Arts and Sciences, Beijing Institute of Fashion Technology, Beijing 100029, China

**Keywords:** multi-USV swarm, path planning, A* algorithm, target hunting, obstacle avoidance

## Abstract

A single unmanned surface combatant (USV) has poor mission execution capability, so the cooperation of multiple unmanned surface ships is widely used. Cooperative hunting is an important aspect of multi USV collaborative research. Therefore, this paper proposed a cooperative hunting method for multi-USV based on the A* algorithm in an environment with obstacles. First, based on the traditional A* algorithm, a path smoothing method based on USV minimum turning radius is proposed. At the same time, the post order traversal recursive algorithm in the binary tree method is used to replace the enumeration algorithm to obtain the optimal path, which improves the efficiency of the A* algorithm. Second, a biomimetic multi USV swarm collaborative hunting method is proposed. Multiple USV clusters simulate the hunting strategy of lions to pre-form on the target’s path, so multiple USV clusters do not require manual formation. During the hunting process, the formation of multiple USV groups is adjusted to limit the movement and turning of the target, thereby reducing the range of activity of the target and improving the effectiveness of the algorithm. To verify the effectiveness of the algorithm, two sets of simulation experiments were conducted. The results show that the algorithm has good performance in path planning and target search.

## 1. Introduction

In the water environment, a single USV has poor ability on task processing and work efficiency. Multi-USV can greatly improve work efficiency, and the multi-USV cooperation has many advantages on exploration ability and task completion ability. Therefore, the research on multi-USV control becomes important [1]. With development of control and communication technology, the multi-USV cooperative hunting has gradually become a hot spot in the field of multi-USV swarm control [2]. The core of the cooperative hunting control is that the multi-USV swarm completes the target hunting task through cooperative control [3]. In recent years, many scholars have studied the multi-USV cooperative hunting control, and many hunting strategies has been proposed. The multi-USV cooperative hunting strategy is generally divided into two parts. First, the path of the USV should be planned. Then, the target is hunted and tracked in real time. In this study, environments with obstacles are defined as water environments with numerous obstacles and constraints. The above constraints include the dynamic features of the target and the smoothness constraints of the planning path.

At present, there are various methods of path planning, which can be divided into global and local path planning based on the cognition of environment information [4]. Global path planning methods can search for an optimal path in a pre-built environment model based on known environment information. Global path planning methods mainly include the A* algorithm [5], Dijkstra algorithm [6], genetic algorithms [7], particle swarm optimization (PSO) [8], and deep reinforcement learning (DRL) algorithms [9]. The above methods usually require long calculation times, and their efficiency is low. Local path planning methods can obtain the surrounding environment’s information by using sensors and autonomously plan a collision-free path in partially unknown environments, so they are suitable for dynamic unknown environments. The commonly used methods mainly include the following three categories. The first type is path planning methods based on virtual potential fields, such as the artificial potential field (APF) method [10]. This kind of method is simple in structure and highly efficient. However, due to the obstacles, it has disadvantages of falling into local minima, and even has the potential to not reach the goal. The second type is path planning methods based on sampling, such as the probabilistic roadmap method (PRM) [11] and rapidly exploring random tree (RRT) [12]. These algorithms have fast sampling speed, but the cost of generating paths is higher, the randomness greatly interferes with the planning rate, and sometimes they cannot even find a feasible solution. Thus, these algorithms are often used to solve high-dimensional planning problems with nonholonomic constraints. The third type is intelligent path planning methods based on bionics, such as ant colony optimization (ACO) [13], the artificial bee colony (ABC) algorithm [14], and the imperialist competitive algorithm (ICA) [15]. Such methods are effective in solving complex optimization problems in low-dimensional, and they are easy to find the global optimal solution. However, the above algorithms are easy to fall into a local optimal solution. Meanwhile, the above algorithms have greater randomness during the iterative process and many parameter adjustments are required. These parameters can only be specified through experience or a neural network. Thus, the algorithm efficiency is low.

Among the above algorithms, the A* algorithm is widely used to solve the path planning problem. But the A* algorithm only selects the optimal path according to the length of the planning path without considering the smoothness of the planning path. As a result, the planning path contains many unnecessary turning points, which is suitable to the motion dynamics of USV. To solve this problem, Zhang et al. proposed a new heuristic function combined with the APF, and introduced it into A* algorithm, which effectively reduces the turning points in the planning path and makes the path smoother [16]. Yu et al. improved the expansion mode of the 8-connections of the traditional A* algorithm to the 20-connection, so that the sharpness of turning at the corner can be greatly reduced and the planning path is smoother [17]. Zhang et al. made a minimum circumscribed circle between two adjacent line segments of the turning point, retains the arc of this segment and removes the sharp corners of the segment to make the path smoother [18]. However, the above algorithm does not take into account that the differences in the movement ability of different USV, which leads to the low generality of the above algorithm. The A* algorithm uses the enumeration algorithm to get the optimal path, and the result is low operation efficiency and a long planning time of the A* algorithm. To solve this problem, Liu et al. combined the Voronoi diagram with the grid method to model the map. As a result, the node does not need to be detected too much, and the total amount of planned paths has been reduced; so the planning time has been reduced [19]. Hong et al. adopted the data structure of the minimum heap and 2D array in the A* algorithm, which reduced the time cost of data processing and reduced the planning time [20]. Zhang et al. introduced a 3D bidirectional sector multilayer variable step search strategy into the A* algorithm to reduce the total number of planning paths. At the same time, the efficiency of the algorithm has improved, and the planning time has reduced [21]. The above methods reduce the planning time by changing the search strategy, but they cannot ensure that the distance of the final planning path is the shortest.

Multi USV collaborative hunting is the key to collaborative control of multi USV populations. Currently, many scholars have studied algorithms related to collaborative hunting with multiple unmanned underwater vehicles. These methods can be divided into two categories: deterministic methods and heuristic methods. The deterministic method uses mathematical tools to solve the virtual capture points and paths of USV. This method has the advantages of low computational complexity, a simple environment, and high efficiency. Representative algorithms include a direct search based on potential field forces [22] and a formation search based on virtual structures [23]. Heuristic hunting by imitating hunting behavior in nature is a method that has a large number of assumptions and can effectively track targets in environments. Representative algorithms include the expulsion ambush method [24] and neural networks based on the hunting method [25]. When the target has strong operability, the expulsion of the AM bus method has high time efficiency, low cost, strong robustness, and fault tolerance [26]. Wang et al. [27] proposed a distributed obstacle avoidance algorithm suitable for multiple mobile robots. This algorithm combines ant colony optimization (ACO) and dynamic window analysis (DWA) to coordinate multiple robot systems through priority strategies, which has high security and global optimality. Guo et al. [28] proposed a distributed collaborative search algorithm and a dynamic target bounding algorithm suitable for quadcopter aircraft clusters, which can effectively search and dynamically monitor dynamic targets in unknown areas. Souza et al. [29] proposed a decentralized multi-agent tracking method using Deep reinforcement learning, and trained a given number of chasers using shared experience. Each agent executes independently at runtime. Although the above algorithms improve the efficiency of target search, they did not achieve preformation before the search. This algorithm has low flexibility. Sun et al. [30] proposed a self-organizing cooperation strategy for multiple unmanned underwater vehicles, dividing the pursuers into a pursuit group and an ambush group based on the escape strategies of evaders under different encirclement states. This method can effectively capture targets and has strong flexibility. Lv et al. [31] assigned tasks based on the multi-layer circular ambush capture model and the characteristics of the pursuit ship. This method can effectively capture targets and has strong flexibility. The above algorithm sets up a pursuit group and an ambush group, which improves the flexibility of the algorithm through collaborative hunting. The above algorithms require high operability for the pursuit group USV, which reduces the practicality of the algorithm.

In summary, there is a problem of unsmooth paths in USV path planning, which is not conducive to USV motion control during sailing. At the same time, most of the existing cooperative hunting algorithms have low efficiency and poor redundancy. Based on the above shortcomings, this paper proposes a cooperative hunting method for multi-USV based on the A* algorithm in an environment with obstacles. First, based on the traditional path planning A* algorithm, a path smoothing method based on the minimum turning radius of USV is proposed. According to the position between obstacles and paths, paths are divided into three scenarios. Second, based on the minimum turning radius of USV, three different methods are used to select new path nodes and then connect these path nodes using ARC. At the same time, the post order traversal recursive algorithm in the Binary tree method is used to replace the enumeration algorithm to obtain the optimal path, which improves the efficiency of the A* algorithm. Finally, in terms of collaborative hunting, a biomimetic multi USV cooperative hunting method was proposed. By simulating the hunting strategy of lion packs, multiple USV packs are preformed along the target’s path and ambushed in a U-shaped array. In the collaborative hunting process, the movement of the target in front of the target is limited by the USV, and the USV on both wings of the target limits the direction of the target’s turn, thereby limiting the range of target activity.

The main contributions of this work can be summarized as follows:(1)A path smoothing method based on USV minimum turning radius was proposed. Based on the A* algorithm, select a new path node and connect it to an arc to make the path smoother. At the same time, the reverse traversal recursive algorithm in the Binary tree method is used to replace the enumeration algorithm to obtain the optimal path, which improves the efficiency of the algorithm and shortens the planning time of the algorithm.(2)A biomimetic based multi-USV collaborative hunting method is proposed. The preformation of multiple USV groups is conducted independently on the path. Multiple USV groups do not require manual formation. The universality of this algorithm has been improved. In the hunting process, the formation of multiple USV groups is adjusted to limit the movement and rotation of the target, effectively reducing the range of target activities, and improving the effectiveness of the algorithm.

This paper is organized as follows. Section 2 provides the basis content of the algorithm theory involved in this paper. Section 3 provides the modeling of this paper. Section 4 presents the algorithm proposed in this paper. The experiment results are discussed in Section 5, and the paper is concluded finally in Section 6.

## 2. Preliminaries

### 2.1. A* Algorithm

The A* algorithm is a classical heuristic search algorithm based on the Dijkstra algorithm combined with the Breadth First Search (BFS) algorithm [32]. The A* algorithm is used to solve the optimal path problem in global path planning. This method continuously approximates the goal by retrieving the nodes, and the optimal path is found. The A* algorithm introduces a heuristic function *h*(*n*) to guide the search direction, omitting searches in irrelevant regions, and thus, it has a high search efficiency. The valuation function of the A* algorithm is defined as follows:(1)f(n)=g(n)+h(n)
where *f*(*n*) is the valuation function from the starting point via node n to the goal point, *g*(*n*) is the actual cost function from the starting point to node n in the state space, and *h*(*n*) is the heuristic estimation cost function of the optimal path from node n to the goal point. The cost function *h*(*n*) is usually expressed in terms of a Euclidean distance as follows:(2)$$h(n)=\sqrt{{\left(x_{n}-x_{goal}\right)}^{2}+{\left(y_{n}-y_{goal}\right)}^{2}}$$
where (*x_n_*, *y_n_*) is the coordinate of node n and (*x_goal_*, *y_goal_*) is the coordinate of the goal.

### 2.2. Binary Tree Method

The binary tree is a tree structure, and the data structure of many problems can be abstracted into the binary tree [33]. The characteristic of the binary tree is that each node can only have two subtrees at most, and the two subtrees can be divided into left and right. The binary tree is a set of n elements which can be composed of empty sets. The set of the binary tree can also be composed of a root and two disjoint subtrees, which can be divided into left and right. It is an ordered binary tree. When the set of binary tree is empty, the binary tree is called an empty binary tree. The element of the binary tree is also called a node. The recursion algorithm of postorder traversal in the binary tree method is a common traversal method of binary tree. The recursion algorithm of postorder traversal in the binary tree method traverses the left subtree at first, then traverses the right subtree, and finally traverses the root. As shown in Figure 1.

## 3. Modeling

### 3.1. USV Modeling

This section provides a mathematical model of the USV, laying the foundation for algorithm research and simulation experiments. The motion coordinate system of the USV on the Earth’s surface is shown in Figure 2. Establish a fixed coordinate system on the Earth’s surface, where the origin *O_e_* can be any point on the Earth’s surface, with the *Y_e_* axis pointing due north and the *X_e_* axis pointing due east, which can be simplified as the {*e*} coordinate system. In practical environments, USVs can heave, sway, surge, yaw, roll, and pitch. This type of USV motion is called a Six degrees of freedom system. To simplify the analysis, this article only considers the yaw, roll, and pitch of the USV. The Six degrees of freedom mathematical model of the USV is simplified to the three degrees of freedom mathematical model of the USV. The three degrees of freedom mathematical model of the USV is represented as:(3)$$\eta^{\prime }=J(\eta )\upsilon$$
where *η* = [*x*, *y*, *ψ*]*^T^* is the pose vector, which defines the position and heading angle; *υ* = [*u*, *v*, *r*]*^T^* is the speed vector, which includes the forward, sway, and steering speeds; *J*(*η*) is the transformation matrix from the Earth coordinate system to the USV motion coordinate system [34].

Modeling a real USV using the Unity3D 2020.3.19f1c2 platform to obtain a 3D model of the USV. The actual USV is shown in the figure. The 3D model of Figure 3a and the USV is shown in Figure 3b. The collision module and collision detection module of the Unity3D platform have been added to the 3D model of the USV to facilitate the observation of collisions in the 3D model of the USV. In this paper, the rigid body and kinematics rigid body collider are added to the 3D model of the USV, so that the 3D model of the USV has inertia and gravity. Finally, add the buoyancy module to the 3D model of the USV. The above modeling methods can enable the 3D model of the USV to have real physical characteristics in simulation experiments, and the proposed algorithm can be more effectively validated in simulation experiments.

### 3.2. Obstacle Modeling

This paper uses the Unity3D platform to model obstacles in the map. Because a USV is navigated only on the horizontal plane, the section of an obstacle on the horizontal plane is regarded as an obstacle in the path-planning process. Owing to the uncertainty of obstacles on the water surface, to ensure the navigation safety of a USV, we must leave a safe distance between a USV and obstacles and conduct the expansion modeling for the obstacles. The obstacle coordinate system {*o*} is established with the center of the obstacle as the origin, as shown in Figure 4.

The boundary coordinates of obstacles are (*x_o.i_*, *y_o.i_*), *i* = 1, 2, …, *n*. The obstacle is expanded, and the expression is as follows:(4)$$\left\{ \begin{array}{l} {\bar{x}}_{o.i}=\left(1+E\right)\cdot x_{o.i}\\ {\bar{y}}_{o.i}=\left(1+E\right)\cdot y_{o.i} \end{array}\right.$$
where *E* is the obstacle expansion coefficient.

### 3.3. Target Modeling

This section is for target modeling, and the motion of the USV on the water surface is shown in Figure 5a. To simplify the analysis, this article imitates the modeling method of USV in Section 3.1 to model the target, considering only the yaw, roll, and pitch of the target, and establishes a mathematical model of a three degrees of freedom target. The target speed is *v_goal_*, and the target angular velocity is *ω_goal_*. This article uses Unity3D to model the target in 3D, as shown in Figure 5b.

## 4. Proposed Algorithm

### 4.1. Improved A* Algorithm

The traditional A* algorithm’s planning path does not comply with the motion constraints of USV, and the turning point of the planning path is sharp. Therefore, it is necessary to smooth the planning path of the A* algorithm to ensure that the USV can navigate on the planning path and reach the endpoint.

*P* is the planning path of the traditional A* algorithm:(5)$$P=[(x_{1},y_{1}),(x_{2},y_{2}),(x_{3},y_{3}),\cdots ,(x_{i},y_{i}),\cdots ,(x_{k},y_{k})]$$
where (*x_i_*, *y_i_*) is the coordinates of the path nodes in the planning path, and *k* is the total number of path nodes of the A* algorithm for path planning based on grid graph method.

Let *k_c_* be the curvature of *P*. *r_max_* is the maximum angular velocity of USV. *u* is the forward speed of the USV. *R* is the minimum turning radius of the USV. The expression is as follows:(6)$$R=\frac{\left|{\dot{\eta }}^{\prime }\right|\cdot (\theta -\varphi )}{2r}$$
where *η*′ and *r* are obtained according to Formula (3). *θ* is the angle between the path *L*′ and the *X_e_*-axis. *φ* is the angle between the path *L* and *X_e_*-axis.

*R_s_* is the safety range of USV. *P* is the set of path nodes before smoothing. *P*′ is the set of smoothed path nodes. (*x_i_*, *y_i_*) is the coordinate of the path node where the USV is located, where *i* = 1, 2, …, *k*. (*x_t_*, *y_t_*) is the coordinate of the turning point. *t* is the abbreviation for *turn*, and (*x_t_*, *y_t_*) is the turning point. (*x_n_*, *y_n_*) is the coordinate of the farthest point of the path section after the truning point, where *n* is the abbreviation for *node*. *L* is the length of path from (*x_t_*, *y_t_*) to (*x_n_*, *y_n_*). *L*′ is the length of path between (*x_i_*, *y_i_*) and (*x_t_*, *y_t_*). As shown in Figure 6, the red straight line represents the local path before smoothing, and the direction of the red arrow is the forward direction of the path.

When the distance between (*x_i_*, *y_i_*) and the (*x_t_*, *y_t_*) is *R_s_* and the curvature of the path *P* is greater than the maximum navigable curvature of the USV, i.e., *L* = *R_s_* and *k_c_* > *r_max_*/*u*, start smoothing the path. If the above conditions are not met, it will be considered a smooth planned path and the original path will be retained. The smoothing method is divided into two steps. First, select the path node based on the relative position between the obstacle and the planning path. Second, the path nodes are connected with arcs for smoothing. The relative position between the obstacle and the planning path can be divided into three categories, and different methods are used to select the path node:

(1) When there is no obstacle within the square with the turning point as the center and the side length of 2*R_s_*, i.e., {(*x*, *y*) ||*x* − *x_t_*| < *R_s_*, |*y* − *y_t_*| < *R_s_*} ∩ *O* = Ø, take the (*x_n_*, *y_n_*) as the *i* + 1-th path node and save the *i* + 1-th path node in *P*′. The coordinates of *x_i_*_+1_ and *y_i_*_+1_ are as follows:(7)$$\left\{ \begin{array}{l} x_{i+1}=x_{i}+\frac{L^{\prime }\cdot \cos\theta +L\cdot \cos\varphi }{\sin(\theta +\varphi )}\\ y_{i+1}=y_{i}+\frac{L^{\prime }\cdot \sin\theta +L\cdot \sin\varphi }{\cos(\theta +\varphi )} \end{array}\right.$$

The result is shown in node 2 on the right in Figure 7.

If there are multiple turning points in the path, the multiple turning points are processed iteratively by (6). In Figure 8, there are two turning points in the planning path. First, re-select the path node according to the first turning point, node 3 on the right side of Figure 8 is obtained by (6) based on node 1. Then, after processing the first turning point, re-select the path node according to the second turning point. Select the point between node 1 and node 3 and away from the second turning point Rs as node 2. Based on node 2, node 4 in the right figure is obtained by (6).

(2) When there are obstacles within the square that the midpoint of path *L*′ as the center and the side length of *R_s_*, i.e., {(*x*, *y*) ||*x* − *x_t_* + (*L* cos*φ*)/2| < *R_s_*/2, |*y* − *y_t_* + (*L* sin*φ*)/2| < *R_s_*/2|} ∩ *O* = Ø, set the turning point as the *i*-th path node. After the *i*-th path node, take the point inclined by 45 degrees along the path direction and the distance is √2*R* as the *i* + 1-th path node, and save the *i* + 1-th path node in *P*′. The coordinate values *x_i_*_+1_ and *y_i_*_+1_ are as follows:(8)$$\left\{ \begin{array}{l} x_{i+1}=x_{i}+R\cdot \tan(\frac{3}{4}\pi -\theta )\cdot \sin(\frac{3}{4}\pi -\theta )\\ y_{i+1}=y_{i}+R\cdot \sec(\frac{3}{4}\pi -\theta )\cdot \cos(\frac{3}{4}\pi -\theta ) \end{array}\right.$$
where *R* is the minimum turning radius of USV. The result is shown in node 3 on the right in Figure 9. Node 2 is the *i*-th path node selected in the first iteration.

After obtaining node 3, set node 3 as the *i*-th path node for the second iteration. After the *i*-th path node, take the point inclined by 30 degrees along the forward direction of the path and the distance is *R* as the *i* + 1-th path node, and save the *i* + 1-th path node in *P′*. The coordinate values *x_i_*_+1_ and *y_i_*_+1_ are as follows:(9)$$\left\{ \begin{array}{l} x_{i+1}=x_{i}+R\cdot \tan(\frac{5}{6}\pi -\theta )\cdot \sin(\frac{5}{6}\pi -\theta )\\ y_{i+1}=y_{i}+R\cdot \sec(\frac{5}{6}\pi -\theta )\cdot \cos(\frac{5}{6}\pi -\theta ) \end{array}\right.$$

The result is shown in node 4 on the right in Figure 9. Then set node 4 as the *i*-th path node, and repeat the above steps to obtain node 5, and save node 5 in *P′*.

(3) When there are obstacles within the square that the midpoint of path *L* as the center and the side length of *R_s_*, i.e., {(*x*, *y*) ||*x* − *x_t_* + (*L* cos *φ*)/2| < *R_s_*/2, |*y* − *y_t_* + (*L* sin *φ*)/2| < *R_s_*/2|} ∩ *O* = Ø. After the *i*-th path node, take the point inclined by 30 degrees along the negative direction of the path after the turning point and the distance is *R* as the *i* + 1-th path node, and save the *i* + 1-th path node in *P*′. The coordinate values *x_i_*_+1_ and *y*_i+1_ are as follows:(10)$$\left\{ \begin{array}{l} x_{i+1}=x_{i}+R\cdot \sec(\frac{5}{6}\pi -\theta )\cdot \cos(\frac{5}{6}\pi -\theta )\\ y_{i+1}=y_{i}+R\cdot \tan(\frac{5}{6}\pi -\theta )\cdot \sin(\frac{5}{6}\pi -\theta ) \end{array}\right.$$

The result is shown in node 2 on the right in Figure 10. Then, set node 2 as the *i*-th path node, and repeat the above steps to obtain node 3, and save the node 3 in *P*′.

Finally, set node 3 as the *i*-th path node. Use (7) to obtain node 4, and save the node 4 in *P*′.

After all the path nodes are obtained, the path nodes need to be connected with arcs for smoothing. Take the path node as the tangent point and the path direction of the path node as the slope to make a straight line:(11)y=Ax+B
where *A* and *B* are the coefficients of the straight line. This straight line is the tangent of the planning path, which determines the values of A and B and can be obtained through differentiation and algebra. In this study, the Unity3D platform can directly obtain tangent information.

Set (*x_i_*_+1_, *y_i_*_+1_) are the coordinates of the *i* + 1-th path node. After obtaining the *i* + 1-th path node, connect the path between (*x_i_*, *y_i_*) and (*x_i_*_+1_, *y_i_*_+1_) using an arc in the circular with radius Ro, as shown in Figure 11.

In Figure 11, the grid represents the path node, and the connection process of the arc path between node 1 and node 4 is shown from (a) to (d). The red line represents the planning path before smoothing. The blue straight line is obtained by (10). The black curve is a smooth path, which is composed of multiple arcs. The expression of the arc is as follows:(12)$$\{\begin{array}{c} {\left(x-x_{c.i}\right)}^{2}+{\left(y-y_{c.i}\right)}^{2}=R_{o}{}^{2}\\ \left|A\cdot x_{c.i}-y_{c.i}+B\right|=R_{o} \end{array},R_{o}\geq R$$
where *x*∈[*x_i_*, *x_i_*_+1_], *y*∈[*y_i_*,*y_i_*_+1_], (*x_c.i_*, *y_c.i_*) is the center coordinate of the arc path from path node *i* to path node *i* + 1. In Figure 11, (*x_c_*_.1_, *y_c_*_.1_) is the center coordinate of the arc path between path nodes 1 to 2, (*x_c_*_.2_, *y_c_*_.2_) is the center coordinate of the arc path between path nodes 2 to 3, and (*x_c_*_.3_, *y_c_*_.3_) is the center coordinate of the arc path between path nodes 3 to 4. After completing path *P*′, replace *P* with *P*′. *R_o_* is the radius of the circle used for smoothing the path. When the coordinates of two path nodes and the tangent equation of the circle are known, the value of *R_o_* can be obtained through Formula (12).

Based on the above content, the planning path of the traditional A* algorithm is smoothed. Using Algorithm 1 for path planning between the starting point and the target point, and the steps of the Algorithm 1 are presented by pseudocode:
**Algorithm 1**: Smoothing method of path turning pointStep1:Initialize parameters;Step2:**If** *L*′ = *R_s_* and *k_c_* > *r_max_*/*u*Step3:**If** {(*x*, *y*) ||*x* − *x_t_*| < *R_s_*, |*y* − *y_t_*| < *R_s_*} ∩ *O* = Ø Step4:Obtain the path node 2 using (6);Step5:**If** {(*x*, *y*) ||*x* − *x_t_* + (*L*′ cos *φ*)/2| < *R_s_*/2, |*y* − *y_t_* + (*L*′ sin *φ*)/2| < *R_s_*/2|} ∩ *O* = ØStep6:Set (*x_t_,y_t_*) as path node 2;Step7:Obtain the path node 3 using (7);Step8:Obtain the path node 4 using (8);Step9:Obtain the path node 5 using (8);Step10:**If** {(*x*, *y*) ||*x* − *x_t_* + (*L* cos *φ*)/2| < *R_s_*/2, |*y* − *y_t_* + (*L* sin *φ*)/2| < *R_s_*/2|} ∩ *O* = ØStep11:Obtain the path node 2 using (9);Step12:Obtain the path node 3 using (9);Step13:Obtain the path node 4 using (7);Step14:Save all path node in *P*′;Step15:Complete the path of *P*′ using (11);Step16:**If** *k_c_* > *r_max_*/*u*Step17:Repeat Step3–16;Step18:**Else** Replace *P* with *P*′;Step19:**Else** Replace *P* with *P*′.

The traditional A* algorithm uses enumeration algorithms to obtain the optimal path. If all possible situations of a certain type of thing are examined one by one and a universal conclusion is drawn, then this conclusion is reliable. This method is called an enumeration method. The enumeration method utilizes the characteristics of fast computing speed and the high accuracy of computers to test all possible scenarios to solve problems and find answers that meet the requirements. Therefore, the characteristic of enumeration is sacrificing time for a comprehensive answer. In the traditional A* algorithm, although the enumeration method can obtain the optimal path by comparing the lengths of all paths, it does a lot of useless work in obtaining the path, resulting in low efficiency of the algorithm. Therefore, it is necessary to solve the problems of low computational efficiency and long planning time in traditional A* algorithms.

In order to obtain the optimal path from all paths, the algorithm uses the post order traversal recursive algorithm in the binary tree method in reference [33] to replace the enumeration method to obtain the optimal path.

### 4.2. A Biomimetic Multi USV Swarm Collaborative Hunting Method

The speed of the target is *v_goal_*. The location of the target is *P_g_* (*x_goal_*, *y_goal_*). The velocity of the *i*-th USV in a multi USV group is *v_ship.i_* (*x_ship.i_*, *y_ship.i_*). For n virtual structural points, there are *n* USVs in a multi-USV swarm. The target path is used to obtain the initial virtual structure point *P_i_* of the USV on the path. The coordinates of *P_i_* are (*x_virtual.i_*, *y_virtual.i_*) and satisfy the following formula:(13)$$\frac{{\left(x_{virtual.i}-x_{goal}\right)}^{2}}{{\left(y_{virtual.i}-y_{ship.i}\right)}^{2}}+\frac{{\left(y_{virtual.i}-y_{goal}\right)}^{2}}{{\left(x_{virtual.i}-x_{ship.i}\right)}^{2}}=\frac{v_{ship.i}}{v_{goal}}$$
where *i* = 1, 2, …, *n*. After obtaining the initial virtual structure points of multi-USV swarm, adjust the position of the virtual structure points on the target path to obtain a U-shaped array. The average value of all initial virtual structural points is *P_a_* (*x_average_*, *y_average_*), which is expressed as follows:(14)$$\{\begin{array}{c} x_{\mathrm{average}}=\sum_{i=1}^{n}\frac{x_{virtual.i}}{n}\\ y_{\mathrm{average}}=\sum_{i=1}^{n}\frac{y_{virtual.i}}{n} \end{array}$$

*P_a_* is the midpoint *P_mid_* of the U-shaped array. *Y_mid_* is the ray from *P_mid_* to *P_g_*. *X_mid_* is a ray that rotates 90 degrees clockwise based on *Y_mid_*. The coordinate system is established with *P_g_* as the origin, *Y_mid_* as the *y*-axis, and *X_mid_* as the *x*-axis. The U-shaped array is shown in Figure 12. The expression for a U-shaped array is as follows:(15)$$y=ax^{2}-b$$
where *a* is the width coefficient, and the larger the value, the wider the U-shaped array, with *a* > 0. *b* is the distance coefficient, and the larger the value, the farther the U-shaped array is from the target, *b* > (4*a*^2^*λ*^2^ + 1)/4*a*. *λ* Is the minimum distance between the U-shaped array and the target.

The U-shaped array is designed to ambush targets. When the target approaches the U-shaped array, multi-USVs can more conveniently hunt the target.

x*_mid._*_1_ is *x_average_*, *y_mid._*_1_ is *y_average_*. *P_mid.i_* is arranged on the U-shaped array, and its *x*-axis coordinate value *x_mid.i_* is as follows:(16)$$x_{mid.i}=x_{mid.i-1}+\frac{v_{ship}\cdot \left(\left|x_{average}-x_{goal}\right|+\left|y_{average}-y_{goal}\right|\right)}{\eta \cdot v_{goal}}$$

The *y*-axis coordinate value *y_mid.i_* is as follows:(17)$$y_{mid.i}=\frac{{\left(x_{mid.i}\right)}^{2}}{a}$$
where *η* is a compaction parameter of the formation, and the larger the value, the more compact the U-shaped array. On the contrary, the smaller the value, the looser the U-shaped array. *i* = 2, …, *n*.

When all USVs reach the formation position, i.e., the distance between all USVs and their corresponding virtual structural points is 0, multi-USV groups begin to pursue the target. The angular velocity of the target is *ω_goal_*, the angular velocity of the *i*-th USV is *ω_ship.i_*. The target is the center of the circle, with a radius of *R_v_*. The expression for radius *R_v_* is as follows:(18)$$R_{v}=\frac{n\cdot \left(v_{goal}\cdot \sum_{i=1}^{n}\omega_{ship.i}+\omega_{goal}\cdot \sum_{i=1}^{n}v_{ship.i}\right)}{\sum_{i=1}^{n}\left(\omega_{ship.i}\cdot v_{ship.i}\right)}$$

The expression of the circle is as follows:(19)$${\left(x-x_{goal}\right)}^{2}+{\left(y-y_{goal}\right)}^{2}=R_{v}^{2}$$

x*_circle._*_1_ is *x_goal_*, the expression of *y_mid._*_1_ is as follows:(20)$$y_{circle.1}=y_{goal}-\sqrt{1-R_{v}^{2}}$$

The virtual structure points on the circle are arranged. The expression of the virtual structure points (*x_circle.i_, y_circle.i_*) is as follows:(21)$$\begin{array}{l} x_{circle.i}=\left\{ \begin{array}{l} x_{circle.(i-1)}+\frac{4R_{v}}{n},1<i\leq \frac{n}{2}\\ x_{circle.(i-1)}-\frac{4R_{v}}{n}-R_{v},\frac{n}{2}<i\leq n \end{array}\right.\\ y_{circle.i}=\left\{ \begin{array}{l} y_{goal}-\left|x_{circle.i}+1-x_{goal}\right|,1<i\leq \frac{n}{2}\\ y_{goal}+\left|x_{circle.i}+1-x_{goal}\right|,\frac{n}{2}<i\leq n \end{array}\right. \end{array}$$

The improved A-star algorithm was used to obtain the planned path from the current position of the USV to the virtual structure point. Then, form a circle around the target, as shown in Figure 13. In Figure 13, the USV directly in front of the target is the intercepting vessel. The blocking ship is used to block the forward escape route of the target. The USVs on both sides of the target are mainly used to limit the target’s turning and be ready to block the target’s escape route at any time.

Narrowing the surrounding circles, multi-USV swarms have achieved hunting. The target is in the center, resulting in a circle with a radius of *R_c_*. The expression for radius *R_c_* is as follows:(22)$$R_{c}=\frac{n\cdot \left(v_{goal}\cdot \sum_{i=1}^{n}\omega_{ship.i}+\omega_{goal}\cdot \sum_{i=1}^{n}v_{ship.i}\right)}{\sum_{i=1}^{n}\left(\omega_{ship.i}\cdot v_{ship.i}\right)}$$

The expression of the circle is as follows:(23)$${\left(x-x_{goal}\right)}^{2}+{\left(y-y_{goal}\right)}^{2}=R_{c}^{2}$$

*x_catch._*_1_ is *x_goal_*, the expression of *y_catch.1_* is as follows:(24)$$y_{catch.1}=y_{goal}-\sqrt{1-R_{c}^{2}}$$

The virtual structural points are evenly arranged in a circle. The position of virtual structural points (*x_catch.i_, y_catch.i_*) is obtained by (20). The improved A-star algorithm was used to obtain the planned path from the current position of the USV to the virtual structure point. Multiple USV groups begin to form a capture circle.

When the target attempts to escape the capture circle, adjust the position of the USV to limit the target and make the actual distance d between the USV and the target close to *R_c_*. The distance *d* between *P_ship.i_* and *P_g_* is as follows:(25)$$d=\left|P_{ship.i}P_{g}\right|=\left|x_{ship.i}-x_{goal}\right|+\left|y_{ship.i}-y_{goal}\right|$$
where *x_ship.i_* is the *x*-axis coordinate of the *i*-th USV. *y_ship.i_* is the *y*-axis coordinate of the *i*-th USV. The USV gathers towards the target direction, reducing the accommodation space between USVs. The escape direction angle of the target is *θ_run_*, the angle between the line from the *i*-th USV to the target and the target direction is *θ_i_*. When the target escapes, adjust the formation of multiple USV groups to block the target’s escape route. The virtual structure point (*x_block.i_, y_block.i_*) of the *i*-th USV needs to satisfy the following expression:(26)$$\left\{ \begin{array}{l} y_{block.i}=\tan\theta_{i}\cdot x_{block.i}-\tan\theta_{i}\cdot x_{goal}+y_{goal}\\ {\left(x_{block.i}-x_{goal}\right)}^{2}+{\left(y_{block.i}-y_{goal}\right)}^{2}=R_{c}^{2}\\ \theta_{i}=\theta_{run}+{\left(-1\right)}^{i}\cdot Dimx[\frac{i+1}{2}]\cdot \frac{\pi }{12} \end{array}\right.$$
where *i* = 1, 2, …, *n*. Finally, the improved A-star algorithm is used to get the planning path from the current position of the USV to the virtual structure point as shown in Figure 14.

## 5. Simulation Experiment and Discussion

This part is mainly about the comparison and analysis of the experimental results. The simulation experiment used Windows 10 as the operating system and Unity3D as the simulation tool. The hardware platform was an Intel Core i5-10200h processor (Intel Corporation, City of Santa Clara, CA, USA) with a main frequency of 2.4 GHz and 16 GB memory. This experiment is implemented in two steps. First, conduct path planning simulation experiments and compare the experimental results. Second, a comparative experiment between the proposed algorithm and the approaching method is implemented to verify the effectiveness of the proposed algorithm. The relevant parameters of the proposed algorithm and simulation experiment in this paper are shown in Table 1, and their values are obtained from reference [8,11].

### 5.1. Simulation Experiment of Path Planning in Scene 1

In Scene 1, the coordinate of the start point is (−55, −284), and the coordinate of the end point is (291, −49). Based on the above map parameters, a comparative experiment between the traditional A* algorithm and the proposed algorithm is conducted to verify the obstacle avoidance ability of the proposed algorithm. The experimental results are shown in Figure 15 and Figure 16.

In Scene 1, the traditional A* algorithm does not consider the path smoothing problem. There are turning points in the planning path that do not account for the motion ability of the USV. As a result, USV deviates from the planning path and collides with obstacles, as shown in Figure 15. The proposed algorithm smooths the planning path based on the minimum turning radius of the USV, so that the USV can navigate on the planning path and reach the end point, as shown in Figure 16. Therefore, the proposed algorithm has stronger obstacle avoidance ability than the traditional A* algorithm, and the planning path of the proposed algorithm is smoother than the planning path of the traditional A* algorithm.

### 5.2. Simulation Experiment of Path Planning in Scene 2

In Scene 2, the coordinate of the start point is (181, −419) and the coordinate of the end point is (−84, 103). The planning path of the traditional A* algorithm is not smooth, the USV deviates from the path and collides with obstacles when tracking the planning path of the traditional A* algorithm. Therefore, in scene 2, based on the above map parameters, the A* algorithm combined with the B-spline curve is selected as the comparison algorithm. The comparison experiment is conducted with the A* algorithm combined with the B-spline curve and the proposed algorithm to verify the effectiveness of the planning path of proposed algorithm and the high operation efficiency of proposed algorithm. The experimental results are shown in Figure 17 and Figure 18, and Table 2.

Combined with the experimental results in Table 2. It can be seen from Figure 17 that the planning time of the A* algorithm combined with the B-spline curve is longer than the planning time of the proposed algorithm, and the length of the planning path in the A* algorithm combined with the B-spline curve is longer than the length of the planning path in the proposed algorithm. And the total turning angle of the A* algorithm combined with the B-spline curve is also larger than the total turning angle of the proposed algorithm. As can be seen from Figure 18, since the algorithm in this paper smooths the path based on the minimum turning radius of the USV, which shortens the length of the planning path in proposed algorithm and reduces the total turning angle in proposed algorithm, the planning path of proposed algorithm is suitable for the motion ability of the USV. In this paper, the recursion algorithm of postorder traversal in the binary tree method is used instead of the enumeration algorithm to get the optimal path, which improves the operation efficiency of the proposed algorithm and shortens the planning time of the proposed algorithm.

In order to avoid the contingency of the experiment results, three groups of different start points and end points are selected in Scene 2 for the simulation experiments. The result is shown in Table 3.

### 5.3. Simulation Experiment of Target Hunting

In order to verify the effectiveness of the proposed algorithm, the approaching method [35] was used to implement comparative experiments with the proposed algorithm. The approaching method is a method to directly plan the path of virtual structure points, and the virtual structure points were generated around the target. The target ship does not escape the hunting of multi-USV swarm in this simulation experiment. The experimental results of the approaching method are shown in Figure 19 and Figure 20, and the experimental results of the proposed algorithm are shown in Figure 21 and Figure 22—where the ordinate d of the curve in Figure 20 and Figure 22 is the actual distance between the USV and the target calculated by Formula (24), and the abscissa is time. In this experiment, the multi-USV swarm consists of three ordinary ships, and the ordinary ships are called OS1, OS2, and OS3.

In Figure 19a, it is shown that the multi-USV swarm was directly approaching the target ship. Although this can reduce the traveling distance of the USV, OS1 was closer to the target ship than OS2 and OS3. In Figure 19b, the result of target hunting is framed by red squares, it is shown that OS1 tracked the target ship and scraped with the target ship. In Figure 20, it is shown that OS2 and OS3 did not effectively encircle the target ship. The distance between the multi-USV swarm was too large, leading to the hunting failure. In Figure 21a, it is shown that the multi-USV swarm had generated a U-shaped array in front of the target ship, making preparations for hunting. In Figure 21b, the result of target hunting is framed by red squares; it is shown that the multi-USV swarm has surrounded the target ship. Because of the U-shaped array, the multi-USV swarm had formed an encirclement circle and successfully hunted the target ship. In Figure 22, it is shown that the USV with the proposed algorithm can maintain a similar distance from the target ship, and the formation of multi-USV swarm is more stable.

In addition, this study utilized the cooperative hunting method based on the Long short-term memory (LSTM) neural network and the cooperative hunting method based on artificial potential field (APF) as a comparative algorithm and conducted 10 comparative experiments with the proposed algorithm. The results are shown in Figure 23.

From Figure 23, the performance of the proposed algorithm is far superior to the cooperative hunting method based on APF. At the same time, the proposed algorithm performs better than the cooperative hunting method based on LSTM in most cases. Although the cooperative hunting method based on LSTM outperformed the proposed algorithm in experiments 3, 8, 10, 15, and 18, it is within a reasonable range. In addition, the cooperative hunting method based on LSTM needs to be trained in advance and then applied to cooperative hunting, while the proposed algorithm does not require training. In summary, the proposed algorithm has high cooperative hunting efficiency.

In order to further verify the effectiveness of the proposed algorithm, escape action is added to the target ship. The target ship will always try to escape from the map boundary. At the same time, when the target ship finds another USV in the environment, it will stay away from the USV in the environment. The experimental results are shown in Figure 24 and Figure 25. Where the ordinate *d* of the curve in Figure 25 is the actual distance between the USV and the target calculated by Formula (24), and the abscissa is time.

In Figure 24a, the start of target hunting is framed by red squares; it is shown that the multi-USV swarm had chased the target ship and formed an encirclement circle to surround the target. In Figure 24b, the process of target hunting is framed by red squares; it is shown that the target ship turned left to escape, and OS1 blocked the escape route of the target ship. In Figure 24c, the process of target hunting is framed by red squares; it is shown that the target ship had performed two escape actions, both of which were blocked by OS1. In Figure 24d, the result of target hunting is framed by red squares; it is shown that the multi-USV swarm had trapped the target ship in the encirclement circle and successfully controlled the target ship to prevent the target ship escaping from the map boundary. Therefore, the proposed algorithm can successfully implement encirclement when facing a target with a certain escape ability. In Figure 25, it is shown that the USV with proposed algorithm can maintain a similar distance from the target ship, and the formation of the multi-USV swarm is stable.

## 6. Conclusions

This paper proposed a cooperative hunting method for multi-USV based on the A* algorithm in an environment with obstacles. First, on the basis of the traditional A* algorithm, a path smoothing method based on the minimum turning radius of USV is proposed. Second, a bionic based cooperative hunting method for multi-USV swarm is proposed to hunt the target. In order to verify the effectiveness of the proposed algorithm, two groups of simulation experiments are implemented. The results show that the proposed algorithm has good effectiveness in path planning and target hunting. The efficiency of cooperative hunting methods based on virtual structural points has been improved, and compared with comparative algorithms, the proposed algorithm can more effectively hunt targets.

This study did not consider the impact of ocean currents on USVs, and the impact of ocean currents will also be considered during the cooperative hunting in future research. In addition, in future research, the proposed algorithm in this paper can be improved in two aspects. First, most of the parameters in this paper are obtained from experience. In the future, the proposed algorithm will be improved with a depth learning algorithm to optimize its parameters. Second, this paper does not study the task allocation method, and the related algorithms will be studied to make reasonable task allocation for the multi-USV swarm in the future.

## Figures and Tables

**Figure 1 sensors-23-07058-f001:**
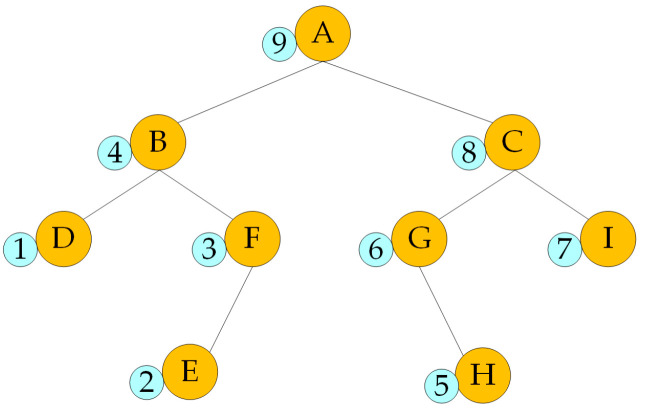
The recursion algorithm of postorder traversal in the binary tree method.

**Figure 2 sensors-23-07058-f002:**
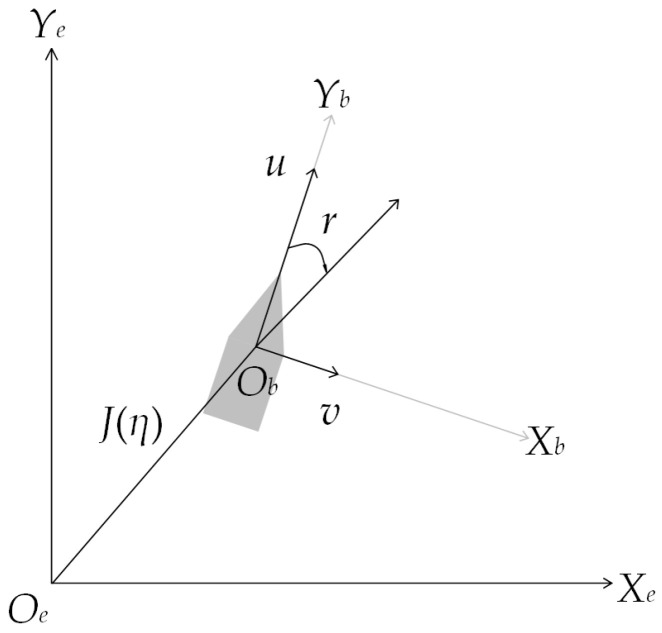
The USV movement coordinate system.

**Figure 3 sensors-23-07058-f003:**
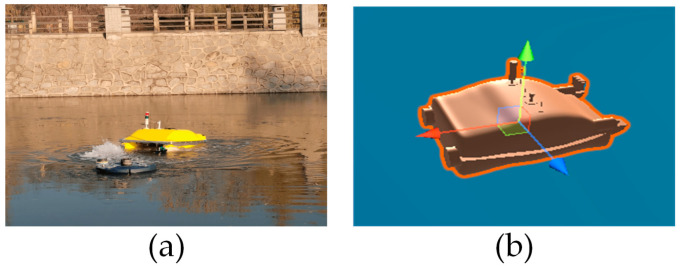
The USV: (**a**) The Real USV; (**b**) The 3D USV.

**Figure 4 sensors-23-07058-f004:**
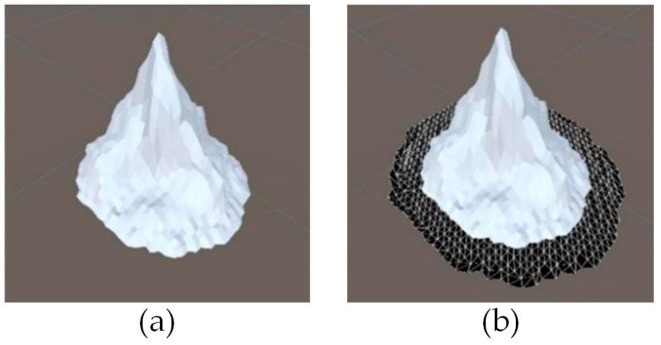
Modeling of obstacle: (**a**) Static obstacle. (**b**) Static obstacle after expansion.

**Figure 5 sensors-23-07058-f005:**
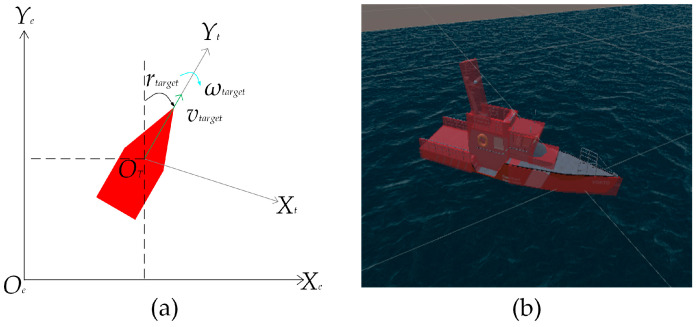
The target: (**a**) Mathematical modeling of target; (**b**) 3D modeling of target.

**Figure 6 sensors-23-07058-f006:**
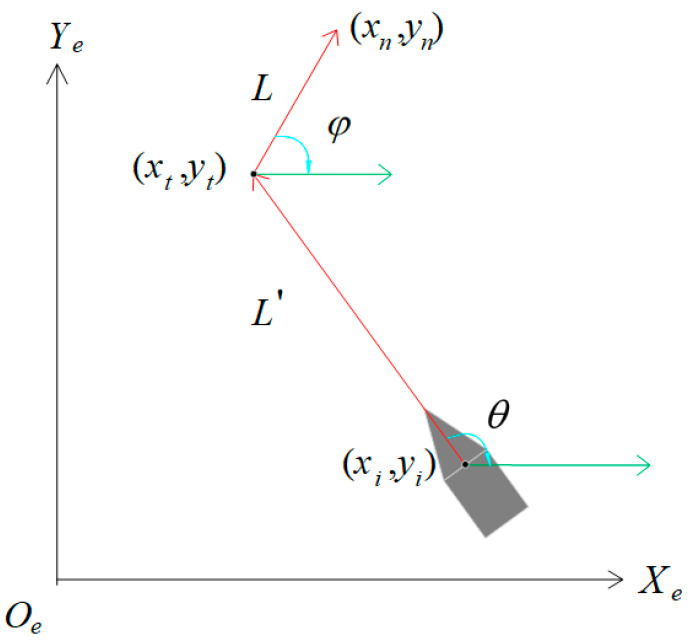
Local path before and after turning point.

**Figure 7 sensors-23-07058-f007:**
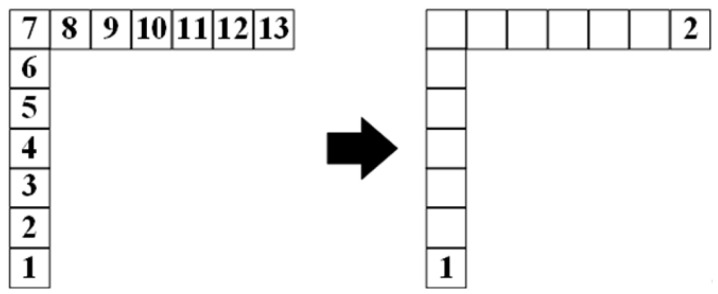
Selection results of the *i* + 1-th path node when there are no obstacles on both sides of the path.

**Figure 8 sensors-23-07058-f008:**
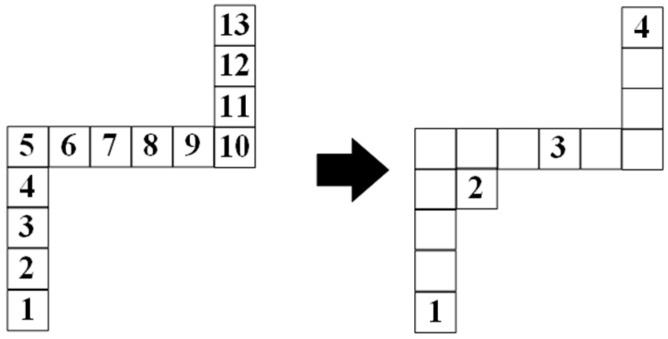
Selection results of path nodes when there are multiple turning points in the path.

**Figure 9 sensors-23-07058-f009:**
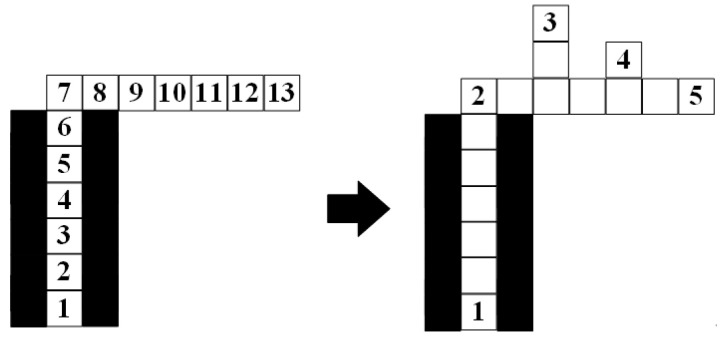
Selection results of path nodes when there are obstacles on both sides of the straight path before the turning point.

**Figure 10 sensors-23-07058-f010:**
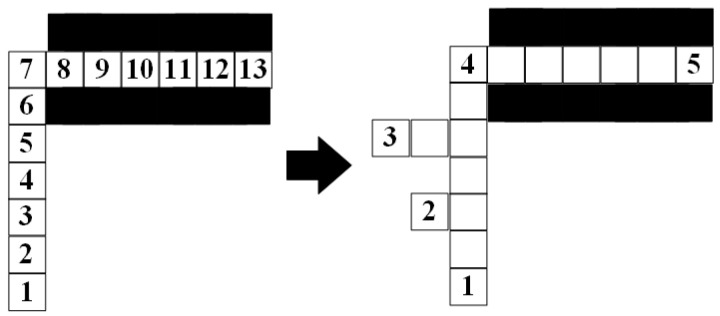
Selection results of path nodes with obstacles on both sides of the straight path after the turning point.

**Figure 11 sensors-23-07058-f011:**
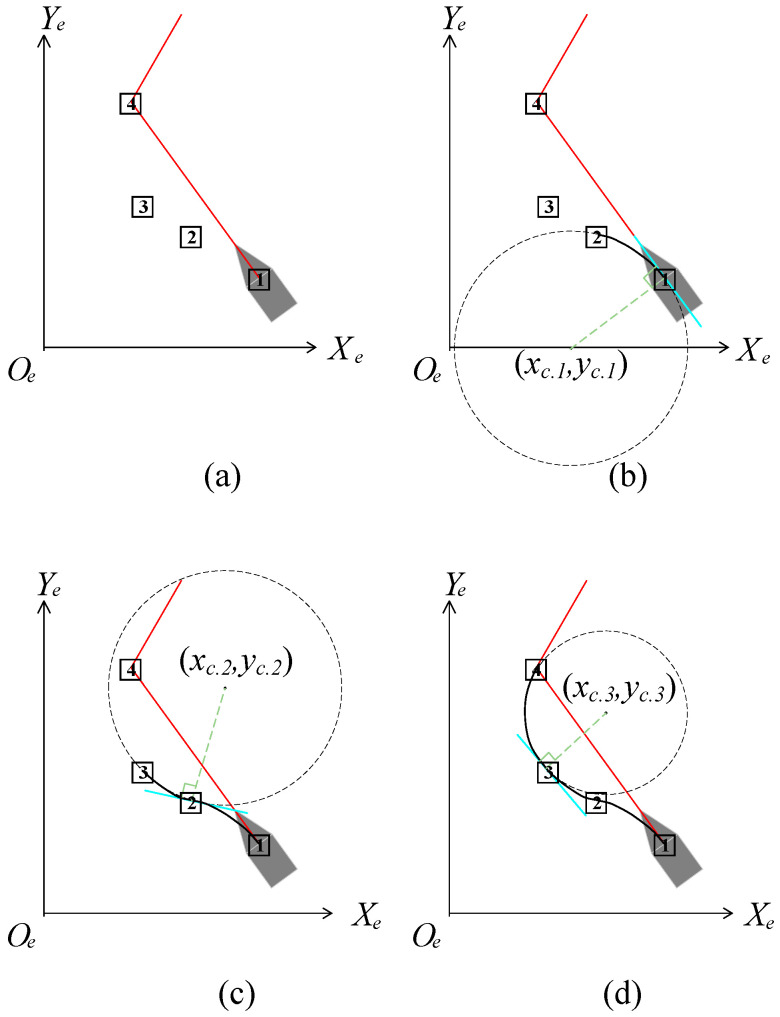
Results of the completing method of the planned path between path nodes: (**a**) before completing; (**b**) connection of nodes 1 to 2; (**c**) connection of nodes 2 to 3; (**d**) connection of nodes 3 to 4.

**Figure 12 sensors-23-07058-f012:**
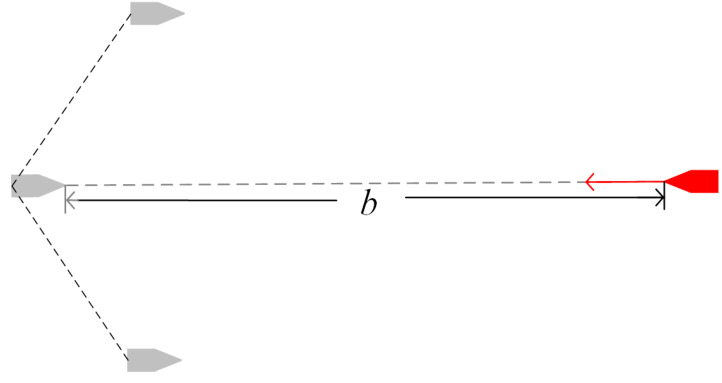
U-shaped array.

**Figure 13 sensors-23-07058-f013:**
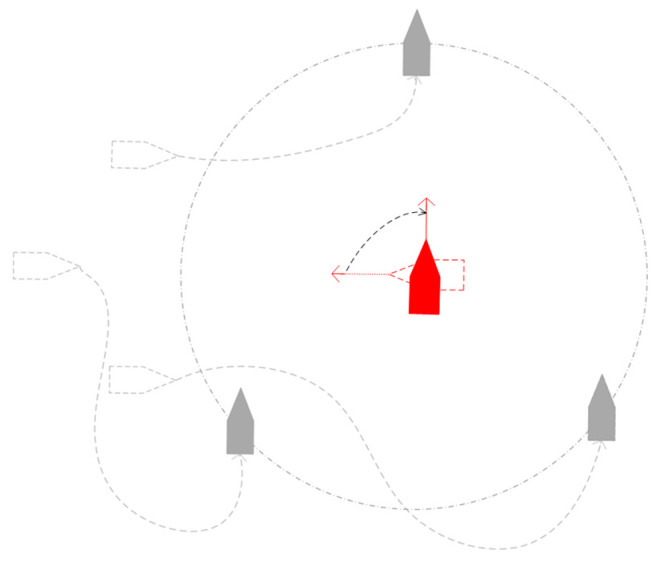
Narrow the surrounding circle for hunting.

**Figure 14 sensors-23-07058-f014:**
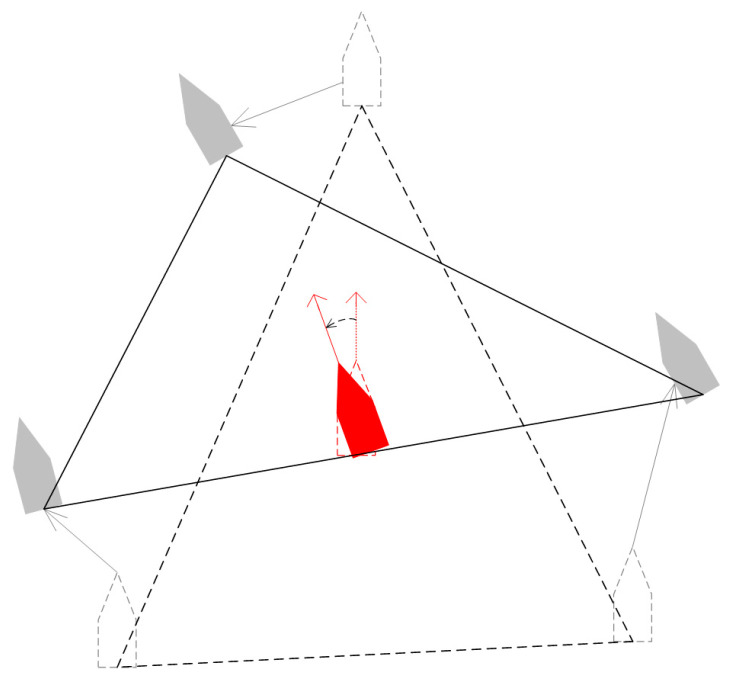
Multi-USV swarm intercepts target.

**Figure 15 sensors-23-07058-f015:**
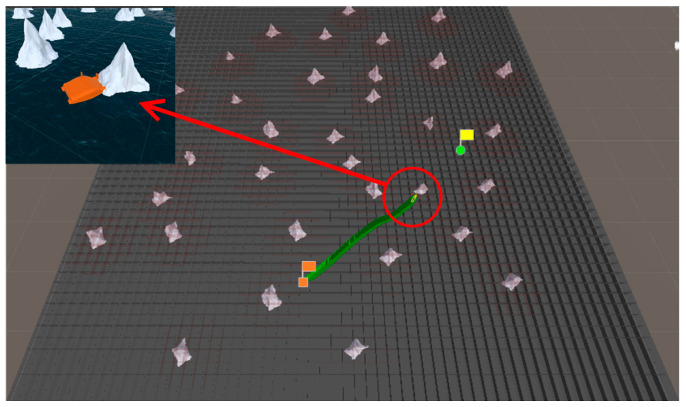
Results of traditional A* algorithm in Scene 1.

**Figure 16 sensors-23-07058-f016:**
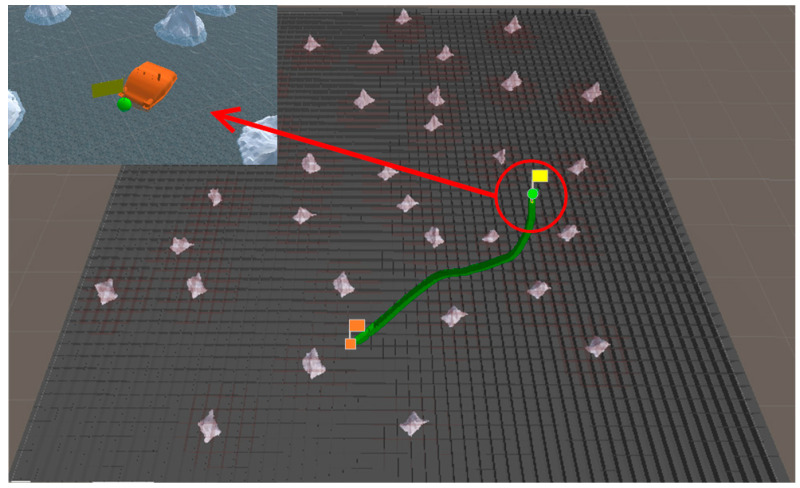
Results of proposed algorithm in Scene 1.

**Figure 17 sensors-23-07058-f017:**
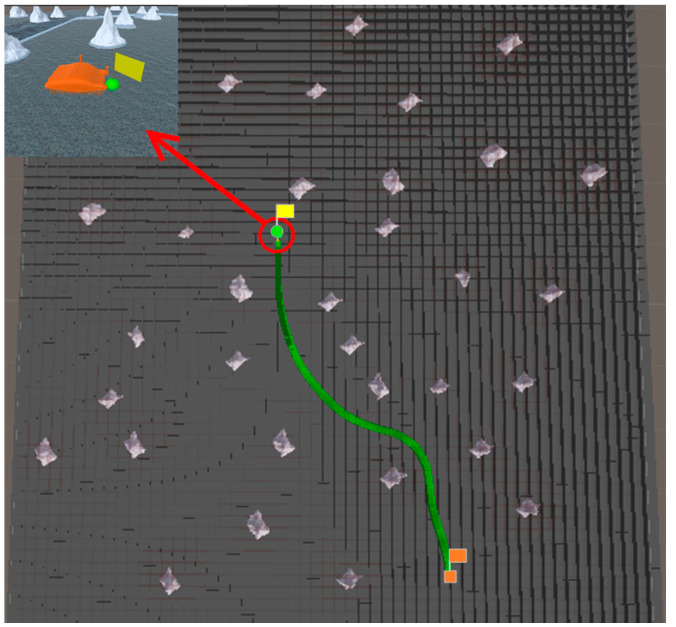
Results of A* algorithm combined with B-spline curve in Scene 2.

**Figure 18 sensors-23-07058-f018:**
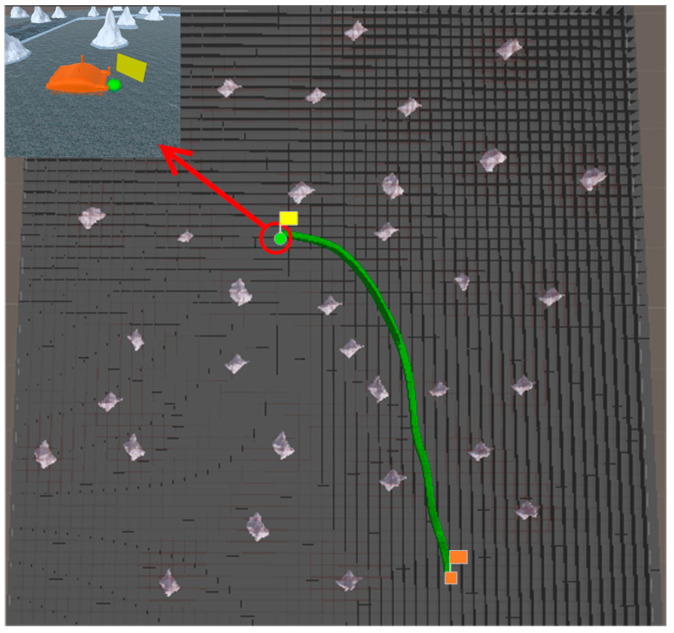
Results of proposed algorithm in Scene 2.

**Figure 19 sensors-23-07058-f019:**
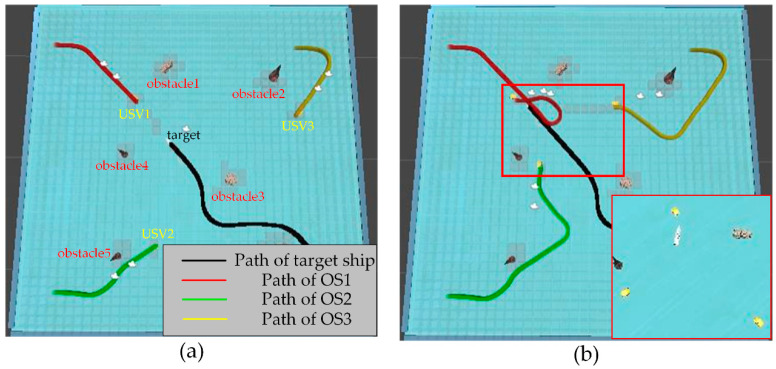
(**a**) The state of approaching method in the process of hunting. (**b**) The result of hunting by approaching method.

**Figure 20 sensors-23-07058-f020:**
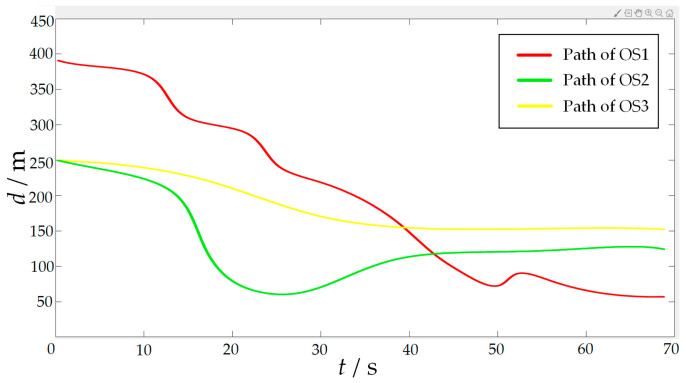
Distance change between the USV and target in the process of surrounding by approaching method.

**Figure 21 sensors-23-07058-f021:**
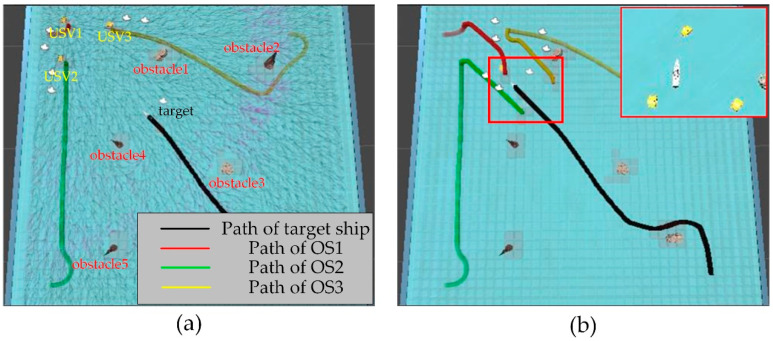
(**a**) The state of proposed algorithm in the process of hunting. (**b**) The result of hunting by proposed algorithm.

**Figure 22 sensors-23-07058-f022:**
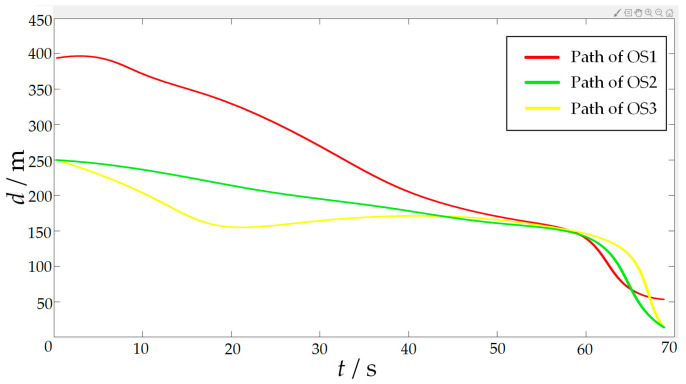
Distance change between USV and target in the process of surrounding by proposed algorithm.

**Figure 23 sensors-23-07058-f023:**
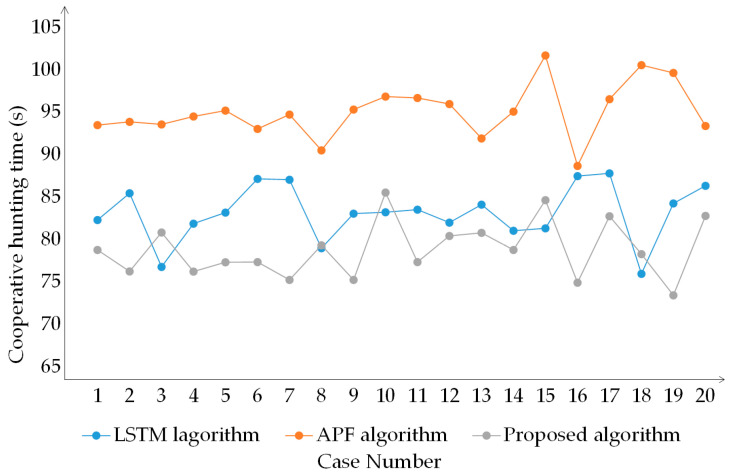
Multiple simulation experimental results of three algorithms.

**Figure 24 sensors-23-07058-f024:**
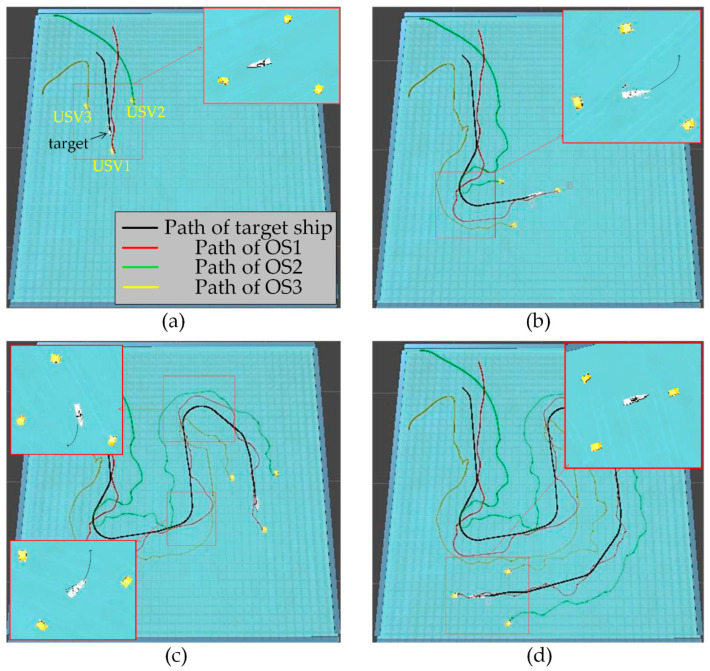
(**a**) One of the results of the hunting at the first moment. (**b**) One of the results of the hunting at the second moment. (**c**) One of the results of the hunting at the third moment. (**d**) One of the results of the hunting at the fourth moment.

**Figure 25 sensors-23-07058-f025:**
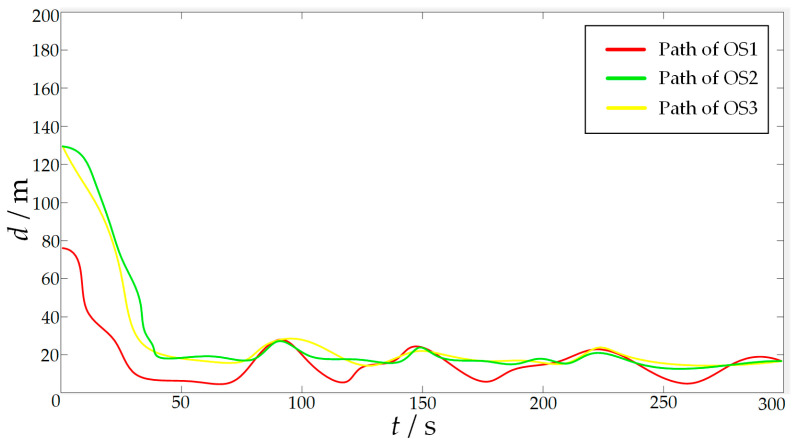
Distance change between the USV and target in the process of hunting by proposed algorithm.

**Table 1 sensors-23-07058-t001:** The algorithm and simulation experimental parameters.

Parameters	Definition	Numerical Value
*u* (m/s)	Forward speed of USV	15
*v_goal_* (m/s)	Forward speed of target	10
*r* (rad/s)	Angular velocity of USV	1
*ω_goal_* (rad/s)	Angular velocity of target	0.7
*E*	Obstacle expansion coefficient	1.5
*a*	Width factor	20
*b*	Distance factor	190
*R* (m)	Minimum turning radius of USV	3
*Rs* (m)	Safety range of USV	6

**Table 2 sensors-23-07058-t002:** Simulation results in Scene 2.

Algorithm Name	Planning Time (s)	Path Length (m)
A* algorithm combined with B-spline curve	0.90211	701.87774
Proposed Algorithm	0.66402	678.60917

**Table 3 sensors-23-07058-t003:** Simulation results of different start and end points under Scene 2.

Start	End	Algorithm	Planning Time (s)	Path Length (m)	Total Turning Angle
(181, −419)	(−84, 103)	A* algorithm combined with B-spline curve	0.90211	701.87774	161°1548′
		Proposed Algorithm	0.66402	678.60917	87°0683′
(−94, −27)	(347, −451)	A* algorithm combined with B-spline curve	0.97951	725.15104	45°1268′
		Proposed Algorithm	0.61845	709.13587	44°1534′
(400, 400)	(−400, −400)	A* algorithm combined with B-spline curve	1.24851	1634.9875	109°8418′
		Proposed Algorithm	0.75112	1568.27831	105°7518′
(400, −400)	(−400, 400)	A* algorithm combined with B-spline curve	1.190011	1612.83788	141°4894′
		Proposed Algorithm	0.74848	1567.15489	134°4537′

## Data Availability

No new data were created.

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
