# Peer review of "A Cooperative Hunting Method for Multi-USV Based on the A* Algorithm in an Environment with Obstacles"

_sensors, 2023, doi:10.3390/s23167058_

Round 1

Reviewer 1 Report

Please refer to the attachment

Reviewer 2 Report

A path smoothing method based on traditional A* algorithm to improve the minimum turning radius of USV is proposed. Additionally, based on this method, a bionic-based cooperative hunting approach for multi-USV swarms is introduced to effectively capture targets. However, the following problems need to be further revised.

1.       In the sentence "Based on the above shortcomings..." in line 126, it refers to the shortcomings identified in previous studies related to cooperative hunting algorithms for multi-USV systems. The author intends to address these shortcomings and improve upon them in their proposed algorithm. However, without further context or specific information from the article, it is difficult to determine the exact aspects that the author has focused on for improvement.

2.       How to understand 'Complex Environment'? It should be clearly defined in the paper.

3.       "3.2 Obstacle modeling" is a three-dimensional obstacle, why can two-dimensional plane coordinates be used? Meanwhile, is this model representative?

4.       Regarding Figure 7, the ranges and meanings of i, t, and n should be supplied.

5.       In line 266, if the condition is not satisfied, how is it handled?

6.       In formulas (7) and (8), what does R represent?

7.       In formula (10), how are A and B assigned values?

8.       Please add titles to figures a, b, c, and d in Figure 12.

9.       In formula (11), what does R0 represent? How is the value of R0 determined or obtained?

10.   Figure 13 is not clear in its description, and it is unclear what issue or concept it is meant to illustrate.

11.   In line 389, "When all USVs reach the formation position..." How is it determined that all USVs have reached the formation position?

12.   Lines 441 to 442 'their values are observed based on experience.' lack authenticity. The sources of these parameters should be clearly stated here

13.   Figure 20, Figure 22, and Figure 24 have poor quality and it is difficult to understand the experimental results. The quality of the figures should be improved, preferably using vector graphics. Additionally, specific information such as obstacles, target ship, USVs, etc., should be clearly labeled in the figures.

A path smoothing method based on traditional A* algorithm to improve the minimum turning radius of USV is proposed. Additionally, based on this method, a bionic-based cooperative hunting approach for multi-USV swarms is introduced to effectively capture targets. However, the following problems need to be further revised.

1.       In the sentence "Based on the above shortcomings..." in line 126, it refers to the shortcomings identified in previous studies related to cooperative hunting algorithms for multi-USV systems. The author intends to address these shortcomings and improve upon them in their proposed algorithm. However, without further context or specific information from the article, it is difficult to determine the exact aspects that the author has focused on for improvement.

2.       How to understand 'Complex Environment'? It should be clearly defined in the paper.

3.       "3.2 Obstacle modeling" is a three-dimensional obstacle, why can two-dimensional plane coordinates be used? Meanwhile, is this model representative?

4.       Regarding Figure 7, the ranges and meanings of i, t, and n should be supplied.

5.       In line 266, if the condition is not satisfied, how is it handled?

6.       In formulas (7) and (8), what does R represent?

7.       In formula (10), how are A and B assigned values?

8.       Please add titles to figures a, b, c, and d in Figure 12.

9.       In formula (11), what does R0 represent? How is the value of R0 determined or obtained?

10.   Figure 13 is not clear in its description, and it is unclear what issue or concept it is meant to illustrate.

11.   In line 389, "When all USVs reach the formation position..." How is it determined that all USVs have reached the formation position?

12.   Lines 441 to 442 'their values are observed based on experience.' lack authenticity. The sources of these parameters should be clearly stated here

13.   Figure 20, Figure 22, and Figure 24 have poor quality and it is difficult to understand the experimental results. The quality of the figures should be improved, preferably using vector graphics. Additionally, specific information such as obstacles, target ship, USVs, etc., should be clearly labeled in the figures.

Reviewer 3 Report

The problem of cooperative hunting for multi-USV is interesting. The state of the art is presented in a good manner and the objectives of this research are adequate. 

The simulation experiment is interesting and the proposed algorithm realize the general objective, but it could be better explained in order to capture the value of scalability.

The conclusions should be extended (line 557). The authors shoul also present the relevant contributions according to the state of the art  in this field of research. At line 558 future work should be related to the limits of this research.

The quality of English is adequate but it could be also improved.

Round 2

Reviewer 1 Report

The revised content is acceptable.

English needs to be improved.

Author Response

We highly appreciate the reviewers' carefulness, conscientious and the broad knowledge on the relevant research fields, since they have given us a number of beneficial suggestions. After a careful study of all comments, we have carefully revised the manuscript and enriched it. Following reviewers' suggestions, the main revisions are as follows:

  1. We have streamlined the abstract of the manuscript and reduced unnecessary wording.
  2. We have corrected the grammar errors in this manuscript. In this manuscript, we highlighted all revisions in yellow.

Reviewer 2 Report

In this new version of the paper, the authors included all the improvements proposed in the previous revision process, so in my opinion it should be considered for publication.

Some minor errors. for example, the "Where" in line 264 should be changed to "where".

Author Response

(The authors gave the same response as above.)
